# RSI-YOLO: Object Detection Method for Remote Sensing Images Based on Improved YOLO

**DOI:** 10.3390/s23146414

**Published:** 2023-07-14

**Authors:** Zhuang Li, Jianhui Yuan, Guixiang Li, Hao Wang, Xingcan Li, Dan Li, Xinhua Wang

**Affiliations:** 1School of Computer Science, Northeast Electric Power University, Jilin 132012, China; liz@neepu.edu.cn (Z.L.); yuanj1anhui@163.com (J.Y.);; 2School of Energy and Power Engineering, Northeast Electric Power University, Jilin 132012, China

**Keywords:** deep learning, object detection, YOLO, remote sensing images

## Abstract

With the continuous development of deep learning technology, object detection has received extensive attention across various computer fields as a fundamental task of computational vision. Effective detection of objects in remote sensing images is a key challenge, owing to their small size and low resolution. In this study, a remote sensing image detection (RSI-YOLO) approach based on the YOLOv5 target detection algorithm is proposed, which has been proven to be one of the most representative and effective algorithms for this task. The channel attention and spatial attention mechanisms are used to strengthen the features fused by the neural network. The multi-scale feature fusion structure of the original network based on a PANet structure is improved to a weighted bidirectional feature pyramid structure to achieve more efficient and richer feature fusion. In addition, a small object detection layer is added, and the loss function is modified to optimise the network model. The experimental results from four remote sensing image datasets, such as DOTA and NWPU-VHR 10, indicate that RSI-YOLO outperforms the original YOLO in terms of detection performance. The proposed RSI-YOLO algorithm demonstrated superior detection performance compared to other classical object detection algorithms, thus validating the effectiveness of the improvements introduced into the YOLOv5 algorithm.

## 1. Introduction

Remote sensing is a technology used to obtain target information by means of remote sensing platforms such as spaceborne and airborne sensors without direct contact with the target. The technology involves comprehensive earth observation with a large amount of target information included in the images, which is widely used in agriculture, environmental monitoring, urban planning, disaster management, and other fields [1,2,3]. With the development and progress of remote sensing technology, remote sensing image observation and acquisition become more efficient. Since objects in remote sensing images are generally small and may be densely distributed, using the human eye to obtain effective information is inefficient and error-prone, and manually extracting this information is an unpractical task. Target detection technology has been widely used in remote sensing [4].

Traditional remote sensing image object detection methods mainly rely on hand-designed features and traditional machine learning algorithms. However, due to the complexity and diversity of remote sensing image data, traditional methods have encountered many difficulties in processing these data. With the rapid development of deep learning technology, especially the appearance of convolutional neural network (CNN), it brings new opportunities and applications for remote sensing image object detection [5,6]. Compared with traditional methods based on manual feature design, deep learning can automatically learn higher-level semantic features through stronger feature representation capabilities and better adapt to complex changes in remote sensing images. Remote sensing image target detection also needs to consider the context information around the target in order to identify and locate the target more accurately. Deep learning models can capture target context information through local receptive fields and pooling operations in convolutional neural networks to improve the accuracy and robustness of target detection. In addition, deep learning models often require large amounts of training data to take advantage of them. Remote sensing image data have rich temporal and spatial information and large-scale coverage, so a lot of training data are provided. This enables deep learning to learn more accurate and generalising models from large-scale data to deal with the diversity and complexity of object detection in remote sensing images.

There are two main types of deep-learning-based object detection algorithms. The first category is the two-stage object detection algorithm, such as spatial pyramid pooling network (SPP-Net) [7], regions with convolutional neural networks (R-CNN) [8], and Faster R-CNN [9]. While these algorithms generally exhibit high detection accuracy, their inherent two-stage nature often leads to bottlenecks in detection speed. The second category of algorithms is the one-stage target detection algorithm, represented by you only look once (YOLO) and single-shot detector (SSD) [10]. These algorithms have fast detection speed but may sacrifice detection accuracy compared to two-stage algorithms. As detection speed is often a high requirement in most complex tasks, one-stage algorithms have advantages in practical applications. This study improved the YOLOv5 algorithm, and several key points of our work are listed as follows:(1)The YOLOv5 algorithm was improved by incorporating a backbone network with an attention mechanism. The design of the attention mechanism was inspired by the human visual system. When people see an image, their gaze remains focused on the parts that they are interested in, with the visual system automatically filtering out the unimportant parts. This is the manifestation of attention, which involves a conscious focus on the important content in an image. From a computational perspective, the attention mechanism enables the algorithm to effectively filter out extraneous information and focus primarily on processing the most relevant information. This study introduces the convolution block attention module (CBAM) [11] attention mechanism into the YOLO object detection algorithm to explore the feasibility of improving the model.(2)The bi-directional feature pyramid network (Bi-FPN) was used to modify the neck section of the network. The FPN structure was proposed to address the challenge of preserving features of smaller objects during the continuous convolution process. This is because larger objects have more pixels, and thus, their features are easier to extract and preserve. In this study, the weighted Bi-FPN of Efficient Det [12], which can support multi-scale fusion more naturally and quickly, was introduced into the neck section of the YOLOv5s network to obtain a more efficient multi-scale fusion method.(3)A small object detection layer was added. When a neural network learns feature information, it can be difficult to learn useful information if the target size is too small. If the downsampling factor is too large, information is easily lost, and if it is too small, extensive GPU resources are required to save feature map information during network forward propagation, considerably reducing the speed of training and inference. In this paper, an additional detection layer was added to the original three detection layers of YOLO to improve the detection performance of small objects.(4)Efficient intersection over union (EIOU) was introduced to improve the loss function. Loss function optimisation is a method of improving the performance of object detection. In the original YOLOv5 loss function, when regressing the predicted values, the effect of the parameter representing the aspect ratio in complete IOU (CIOU) loss [13] is reduced if the aspect ratios of the predicted and ground truth boxes are linear. This study introduces EIOU loss [14] to replace the original CIOU loss and improve the training performance of the network.

## 2. Related Work

Deep learning has achieved remarkable performance in various computer vision and remote sensing tasks [15]. Small object size is one of the important challenges for target detection in remote sensing images. Numerous researchers have used object detection algorithms based on deep learning to propose improvements for this difficulty. Jiarong Ma et al. [16] improved on Faster R-CNN and proposed a small- and medium-sized target recognition for herbivores in large-scale images. This method uses HRNet feature extraction network to detect small targets better and proposes a new overlapping segmentation method to solve the problem of target misfit and missing detection. Wenke Wang et al. [17] proposed an extraction method of traditional village buildings. This method improves Mask R-CNN to use the Path Aggregate Feature Pyramid Network (PAFPN) and Atlas Space Pyramid Pool (ASPP) to enhance the backbone model for multi-scale feature extraction and fusion. On the basis of Mask R-CNN, Qifan Wu et al. [18] used the improved SC-conv based on the ResNet101 backbone network to obtain more discrimination feature information and modified the convolution size to propose an aircraft detection method called SCMask R-CNN in remote sensing images. The detection effect of this method on aircraft target in the DOTA dataset is improved compared with the baseline training algorithm.

All the algorithms proposed by the above researchers are two-stage target detection algorithms, which show a slow detection speed. Bihan Huo et al. [19] proposed SAFF-SSD, a Self-Attention Combined Feature Fusion-Based SSD for Small Object Detection in Remote Sensing. On the basis of SSD, SAFF-SSD improves the feature extraction capability by proposing a transformer-based attention module called the Local Lighted Transformer block. It also uses the CSP-PAN topology as a detection neck. Alessandro Betti et al. [20] proposed a YOLO-like network for small target detection in aerial imagery called YOLO-S. It utilises a small feature extractor and a reshape-passthrough layer to facilitate feature reuse across the network and combine low-level location information with more meaningful high-level information. This method improves the detection performance of small targets. Huaqing Lai et al. [21] designed a feature extraction module combining convolutional neural networks (CNN) and multi-head attention to obtain a larger receptive field and proposed STC-YOLO. The normalised Gaussian Wasserstein distance (NWD) metric is also introduced to improve the sensitivity of the algorithm loss to the position deviation of small targets. The algorithm achieves good results in traffic sign detection. Wang Jian et al. [22] analysed the problems of high resolution and complex background of UAV aerial images and proposed MFP-YOLO, a lightweight detection algorithm based on YOLOv5. This method combines a multi-path inverse residual module and attention module, as well as using a parallel deconvolution space pyramid pool to extract scale-specific information to improve the detection performance of the algorithm.

The aforementioned studies have introduced various improvements to the target detection algorithm and promoted the development of target detection in remote sensing images. In summary, it is necessary to balance the relationship between detection accuracy and computational speed when selecting an algorithm for improvement. In order to improve the detection accuracy without reducing the detection speed too much, this study further improves the one-stage target detection algorithm YOLOv5.

## 3. YOLOv5 Object Detection Algorithm

YOLOv5, which has a small network structure and more convenient configuration, can efficiently handle object detection tasks. The network structure has four variants, namely, YOLOv5x, YOLOv5l, YOLOv5m, and YOLOv5s, in order of decreasing number of parameters. The main difference between these structures is the depth of the model and the number of channels in the convolutional layers; the overall architecture is similar. This study used the YOLOv5s structure, which consists of four parts: input, backbone, neck, and head, as shown in Figure 1. Each part is introduced in detail in the following sections.

### 3.1. Input

The input module normalises images of different sizes and converts them into a tensor of 640 × 640 × 3 for input into the network, thereby generating initial prediction boxes, using the anchor box mechanism. YOLOv5 provides three sets of pre-set anchor box sizes. If the initial anchor boxes and the target size of the dataset do not satisfy certain conditions, the program uses k-means [23] and genetic evolution algorithms to determine the anchor box size that best matches the target of the current dataset.

### 3.2. Backbone

The backbone module is responsible for feature extraction and is designed based on CSP-Darknet53. It consists of four modules: Conv, Focus, C3, and SPP [24]. Each of these modules is described below.

Figure 2 shows the Conv module, which is composed of a convolution operation and the Conv structure of the convolution operation, the BN (batch normalisation) [25] structure, and the SiLU activation function, so the module can also be called the CBL module. As the basic module in the YOLOv5 network, this module constitutes all convolution operations in the network.

Figure 3 shows the Focus module. The design idea of this module is to slice images in a way similar to subsampling and then splicing them together. In this way, the width and height information of the original image is segmented and aggregated into the channel. This can reduce the amount of computation and improve the training speed of the network.

Figure 4 shows the C3 module, which is equal to a simplified version of Bottleneck. It consists of Bottleneck and three Conv structures. Bottleneck Bool aims to further extract and enhance characteristic information. The C3 module is designed to improve feature representation and deepen the layers of the network without increasing the amount of computation.

Figure 5 shows the SPP module, which first halves the input channel through a standard convolution module, then performs maxpool operations of different kernel sizes, and finally splicing so that more features of different sizes can be fused and more network information can be obtained before it is sent to the Neck.

### 3.3. Neck

The Neck module can further enhance the diversity and robustness of features. While YOLOv5 has undergone some minor adjustments compared to its predecessor, it still incorporates the FPN [26] + path aggregation network (PAN) [27] for structural design. The upsample module performs normal upsampling operations, and the concat module concatenates tensors along a certain dimension, which is not equivalent to an add operation. The add operation adds corresponding elements of tensors with the same size, without changing the size of the tensor. However, the concat module merges two tensors into one, and during this process, the size of the resulting tensor may change. The remaining modules are same as those mentioned in the backbone module.

### 3.4. Head

The head outputs the detection results of the network. It consists of three detection modules for large, medium, and small objects. The most important part of the head is the loss function. For machine learning algorithms, learning refers to autonomously improving the efficiency of the model to complete the task, and the loss function can measure the degree of excellence of the model and then quantify the effectiveness of the model. The loss function is also often called the objective function, and the machine learning task hopes to optimise the function to the lowest point. The YOLOv5 loss function is expressed in Equation (1) and consists of classification, objectness, and localisation losses, where the various $\lambda_{i}$ are the balancing coefficients. In the literature, localisation and objectness losses are sometimes referred to as bounding box and confidence losses, respectively.
(1)$$L\mathrm{oss}=\lambda_{1}L_{\mathrm{cls}}+\lambda_{2}L_{obj}+\lambda_{3}L_{\mathrm{loc}}$$

In traditional simple classification tasks, class labels are mutually exclusive. For example, a target could be a chicken, a duck, or even a goose, but it can only belong to one of these classes. Therefore, the softmax function is often used to convert predicted values into probability values that add up to 1, with the highest value constituting the result. In YOLOv3 and later versions, the algorithm considers the possibility that a target may belong to multiple classes. For example, when recognising a person, it may recognise “child” and “running” as two results. In this case, the probabilities of each class are treated independently, and the classification loss function utilises cross-entropy (BCE) loss [28], which is calculated using Equation (2). Here, *N* represents the total number of classes; $C_{ij}$ is the true class value; ${\hat{C}}_{ij}$ is the class value after activation; and $x_{i}$ is the current class prediction value.
(2)$$\begin{array}{l} L_{cls}=-\sum_{n=1}^{N}\left(C_{ij}\ln({\hat{C}}_{ij})+(1-C_{ij})\ln(1-{\hat{C}}_{ij}\right))\\ {\hat{C}}_{ij}=Sigmoid(C_{ij})=1+e^{-x_{i}} \end{array}$$

As part of object loss calculation, BCE loss is calculated using Equation (3). Here, $C_{ij}$ represents the true value of the presence of an object, where 0 represents presence and 1 represents absence. ${\hat{C}}_{ij}$ represents the probability of the presence of an object after the activation function, and its calculation is the same as that of the sigmoid function in the classification loss equation.
(3)$$L_{obj}=-\sum_{n=1}^{N}\left(C_{i}\ln({\hat{C}}_{i})+(1-C_{i})\ln(1-{\hat{C}}_{i}\right))$$

Localisation loss is calculated using *CIOU* loss, which considers three geometric parameters in object detection tasks: overlap area, distance ratio of centre points, and aspect ratio. As shown in Equation (4), the value of localisation loss is equal to 1 minus the value of *CIOU*, where $\rho^{2}(b,b^{gt})$ represents the Euclidean distance between the centre points of the predicted box and the true box; *c* represents the diagonal distance of the minimum closed bounding box that contains both the true and predicted boxes; υ is a parameter used to measure the aspect ratio; and α is a weight function.
(4)$$\begin{array}{l} L_{\mathrm{loc}}=1-CIoU\\ CI\mathrm{oU}=\mathrm{IoU}-(\frac{\rho^{2}(b,b^{gt})}{c^{2}}+\alpha \upsilon )\\ \upsilon =\frac{4}{\pi^{2}}{\left(\arctan\frac{w^{gt}}{h^{gt}}-\arctan\frac{w}{h}\right)}^{2}\\ \alpha =\frac{\upsilon }{(1-IoU)+\upsilon } \end{array}$$

Regarding the loss function, there are two points to consider. First, the target loss uses three different weights to calculate the final result for the three prediction feature maps with different sized objects. These weights, denoted by the parameters in front of each prediction feature map in Equation (5), are hyperparameters that are set based on the common objects in context (COCO) [29] dataset. Second, not all losses calculate the loss for all samples. For instance, the classification and localisation losses only calculate the loss for positive samples, while the target loss, which is based on the centre-intersection over union (*CIOU*) of the target bounding box, needs to be calculated for all samples.
(5)$$L_{obj}=4.0*L_{obj}^{small}+1.0*L_{obj}^{medium}+0.4*L_{obj}^{l\arg e}$$

The YOLOv5 algorithm uses the PyTorch framework, which is easier to deploy compared to the previous Darknet framework. In this study, this version of the YOLO object detection algorithm was used for development and improvement, thereby evaluating the effectiveness of algorithmic improvements through experiments.

## 4. Improved Methods of YOLOv5s

### 4.1. Introducing Attention Mechanism

The attention mechanism enables neural networks to focus on important features and ignore unimportant ones [30]. Convolutional operations extract features by mixing channel and spatial information so that the design of attention mechanisms focus on both of these aspects. CBAM consists of two parts: a channel attention module (CAM) and a spatial attention module (SAM), as shown in Figure 6. By redistributing weights, these two modules can help the network better learn category features and location information of target objects.

The channel attention module focuses on the category of the input image target, as shown in Figure 7. The module takes the input feature map and applies global max pooling and global average pooling to obtain two feature maps. These two 1 × 1 × C feature maps are then fed into a multilayer perceptron (MLP) [31] with a fully connected neural network of two layers. The first layer has C/r neurons, where r is the reduction ratio, and employs a leaky ReLU as the activation function. The second layer consists of C neurons, and the two-layer neural network constitutes a shared fully connected layer. The output features of the shared layer are added together, and a sigmoid normalisation operation is applied to generate the final channel attention features.

This process is expressed mathematically in Equation (6), where $M_{c}$ generates the feature; AvgPool and MaxPool are global average pooling and global maximum pooling operations, respectively; σ is the sigmoid function; and $W_{1}$ and $W_{0}$ are the shared parameters of the MLP network.
(6)$$M_{c}(F)=\sigma (MLP(AvgPool(F))+MLP(MaxPool(F)))=\sigma (W_{1}(W_{0}(F_{avg}^{c})+W_{1}(W_{0}(F_{\max}^{c})))$$

The spatial attention module focuses on the position of the input image target, as shown in Figure 8. The module accepts the feature map output of the channel attention module as the input feature map. First, the feature map is passed through a global max pooling layer and a global average pooling layer, which are then concatenated along the channel axis to obtain a single feature map. This feature map is then reduced in dimensionality to a single channel, using a convolutional operation. The sigmoid normalisation operation is then applied to generate the final spatial attention feature.

This process is expressed mathematically in Equation (7), where $M_{s}$ generates the feature, and $f^{7*7}$ is a convolutional operation with a kernel size of 7 × 7.
(7)$$M_{s}(F)=\sigma (f^{7*7}(Concat(AvgPool(F),MaxPool(F))))=\sigma (f^{7*7}(Concat(F_{avg}^{s},F_{\max}^{s})))$$

The final output feature is obtained by multiplying the feature output of the spatial attention module with the input feature. The newly generated feature map can improve the connection between various features in both channel and spatial dimensions, making it more effective in extracting the relevant features of the target.

### 4.2. Feature Fusion Structure Improvement

The neck section of YOLOv5 network adopts the FPN+PAN structure. The higher layers of the convolutional neural network are more sensitive to semantic features; however, because of the smaller feature map size, there are fewer position features, which is not conducive to detecting target location. The lower layers of the network are more sensitive to position features because of the larger feature map size; however, there are fewer semantic features, which is not conducive to detecting the target category. FPN is a structure that integrates deep and shallow features. The structure diagram of FPN is shown in the blue dotted box of Figure 6. FPN first processes feature maps bottom-up and generates a pyramid-like structure by sending them to a pre-trained network. Then, in a top-down manner, it first duplicates the high-level feature maps and then enlarges the feature map size by upsampling. Subsequently, it reduces the dimensions of previously processed specific feature maps and performs element-wise addition to obtain a new layer. The FPN structure allows the network to obtain better semantic features, whereas the network features allow the PAN structure to better transmit position information. The PAN structure is shown in the red dotted box of Figure 9. PAN first duplicates the lowest layer of the feature pyramid, performs downsampling, and concatenates the output with the second layer of the FPN pyramid to fuse features, which is essentially the reverse operation of the FPN structure.

The PAN structure adopted by YOLOv5 is based on the feature fusion idea of the original PANet structure. Unlike PANet, which uses pixel summation (add operation), YOLOv5 uses channel concatenation (concat operation) to fuse features. In this project, the weighted bidirectional feature pyramid network structure of Bi-FPN is used to replace the PAN structure of the YOLOv5 network. PANet does not set weights for fusing features of different scales, whereas Bi-FPN introduces weights to balance the information in the features of different scales. The Bi-FPN structure is shown in Figure 10, where Pi represents the feature map obtained from the backbone network. By adding the Bi-FPN structure, the algorithm achieves more efficient and richer feature fusion.

### 4.3. Adding the Small Object Detection Layer

Detecting small objects has always been a challenging problem in the field of object detection. The small number of pixels occupied by small objects in the image limit the size of the receptive field, making it difficult to learn the features of the target. If the image size is large, data information is easily lost during high magnification downsampling, and when downsampling at a low magnification, the network needs to retain a large amount of feature information in memory, which can easily cause interruption of the detection task because of resource occupation issues on the graphics card. Utilising the feature extraction process of the YOLOv5 algorithm shown in Figure 11, this study improves the algorithm’s ability to recognise small objects by adding a small object detection layer.

The network structure of this study, with the small object detection layer added to the original YOLOv5 network model, is shown in Figure 12. The modified part is highlighted by a green dashed box. The upsampling operation after the 17th layer of the network enlarges the size of the feature map, making it the same size as the feature map generated by the second layer of the backbone network, which is 160 × 160. This feature map size has a relatively small receptive field. Then, the feature maps from these two layers are concatenated using the concat operation to obtain a larger feature map. Finally, the detection layers in the original network, which use the 17th, 20th, and 23rd layers, are modified so that the 21st, 24th, 27th, and 30th layers are used as detection layers. These modifications add a new detection layer to the network for small object detection.

Adding detection layers increases the computational complexity of the network and consequently slows down the inference speed of the model. However, during training, recall rate is improved, and target loss is significantly reduced. 

### 4.4. Improvement of Loss Function

Improving the loss function is a common approach to optimise object detection models. As mentioned earlier, YOLOv5 uses CIOU loss as the localisation loss function to represent the width and height losses of the predicted box in terms of aspect ratio. However, aspect ratio may not accurately represent the true width and height. If there is a linear relationship between the true and predicted values of the aspect ratio, the role of parameter υ is insignificant. According to the formula, the width and height of the predicted box are inversely proportional to each other. This means that when one increases, the other decreases, making it difficult for the model to optimise both dimensions effectively. In this project, *EIOU* loss is used to solve the problem of CIOU loss, and its formula is shown, as expressed in Equation (8).
(8)$$L_{EIoU}=L_{IoU}+L_{dis}+L_{asp}=1-IoU+\frac{\rho^{2}(b,b^{gt})}{c^{2}}+\frac{\rho^{2}(w,w^{gt})}{C_{w}^{2}}+\frac{\rho^{2}(h,h^{gt})}{C_{h}^{2}}$$

The *EIOU* loss consists of three parts: the overlap loss between predicted and true boxes, centre distance loss, and width and height loss. The first two components are identical to the CIOU loss, but the third loss improves the aspect ratio by treating the difference between the predicted and true box width and height separately. The $C_{w}^{2}$ and $C_{h}^{2}$ in the formula represent the width and height of the minimum bounding rectangle of the predicted and true boxes, respectively. The iterative process of the predicted box regression using CIOU and *EIOU* losses is compared in Figure 13, where it is observed that the width and height of the predicted box cannot increase or decrease simultaneously using CIOU loss; however, this is possible with *EIOU* loss. The *EIOU* loss function improves the regression accuracy of the predicted box, which in turn helps the model to converge faster during the training process.

## 5. Experiment and Results Analysis

### 5.1. Experimental Environment and Parameter Settings

This project was developed using Python, and the detection task was performed on a Windows 10 operating system. However, existing Windows 10 computers cannot meet the minimum requirements of the high-performance training task. Therefore, the training task of this project was implemented by renting a cloud server provided by Auto DL. The specific experimental environment configuration is shown in Table 1.

The project primarily used the YOLOv5 algorithm to complete the experimental task, selecting the YOLOv5s pre-trained on the COCO dataset provided by the official site as the training model. The specific experimental parameter settings are shown in Table 2.

### 5.2. Evaluation Metrics

In object detection tasks, recall, precision, and average precision are commonly used as evaluation metrics. The basic elements of these evaluation metrics are shown in Table 3.

(1)*Recall* represents the proportion of correctly detected targets to the total number of targets, as shown in Equation (9).


(9)
$$Recall=\frac{TP}{TP+FN}$$


(2)*Precision* represents the proportion of correctly detected targets to the total number of predicted targets, as shown in Equation (10).


(10)
$$Precision=\frac{TP}{TP+FP}$$


*Average precision* (AP) represents the area enclosed by the P-R curve, which has recall and precision as the x and y axes, respectively. The formula for calculating the average precision is shown in Equation (11). To obtain the mean average precision (mAP) for multiple categories, the AP values of each category were averaged. In this project, mAP@0.5 and mAP@0.5:0.95 were used as the main evaluation metrics for precision; mAP@0.5 represents the average precision for an IOU of 0.5, whereas mAP@0.5:0.95 represents the average precision for IOUs ranging from 0.5 to 0.95, in increments of 0.05.
(11)$$AP=\int_{0}^{1}P(r)dr$$

For convenience, mAP@0.5 is referred to as $AP_{0.5}$ and mAP@0.5:0.95 is referred to as $AP_{0.95}^{0.5}$ in the remainder of this paper.

### 5.3. Experimental Results Analysis

1.DOTA Dataset

The DOTA [32] dataset was generated from Google Earth, as well as JL-1 and GF-2 satellite images. It includes fifteen categories: airplanes, ships, storage tanks, baseball fields, tennis courts, basketball courts, athletic fields, harbours, bridges, large vehicles, small vehicles, helicopters, roundabouts, soccer fields, and swimming pools. The original images in the dataset were too large for this project, as they caused high GPU utilisation during training. Therefore, the images were cropped to a size of 1024 × 1024. The final dataset for this project contained 21,046 images.

RSI-YOLOv5 was trained on the DOTA dataset in this project. The training took a total of 14 h and 31 min. The $AP_{0.95}^{0.5}$ and $AP_{0.5}$ scores for all categories in the improved YOLOv5 algorithm were 45.3% and 71.2%, respectively. The detection results are shown in Table 4 for all categories in the dataset.

The project compared the RSI-YOLO with the original YOLOv5s algorithm and selected several representative target categories. The results are shown in Table 5, demonstrating that the RSI-YOLO achieved better performance on most metrics. Specifically, for the “airplane” category, the recall rate increased by 1.2%, and the $AP_{0.5}$ and $AP_{0.95}^{0.5}$ scores increased by 1.1% and 2.3%, respectively. For the “ship” category, the recall rate increased by 2.2%, and the $AP_{0.5}$ and $AP_{0.95}^{0.5}$ scores increased by 1.3% and 2.8%, respectively. For the “small vehicle” category, the recall rate increased by 5.6%, and $AP_{0.95}^{0.5}$ increased by 2.5%. Overall, the recall rate for all categories increased by 4.6%; $AP_{0.5}$ and $AP_{0.95}^{0.5}$ increased by 1.3% and 1.6%, respectively.

For remote sensing image object detection tasks, the targets are often located far away from the detection devices, and they may move slowly within the detection field of view. Therefore, it is necessary to optimise detection accuracy and other parameters of the algorithm to accurately detect the targets and their speed. The experimental results demonstrated that the improved algorithm proposed in this project for YOLOv5s effectively improved detection performance for typical small object detection tasks, such as remote sensing image object detection.

Figure 14 presents the detection results for several test images, demonstrating that RSI-YOLO effectively detected most targets in the dataset, including small objects such as vehicles, airplanes, and ships, as well as other targets such as baseball fields, basketball courts, and bridges.

In the second column of Figure 11, RSI-YOLO detected two airplane targets located in the upper right of the image, which were missed by the original YOLOv5 algorithm, indicating the robustness of the proposed algorithm. There was a similar situation in the result presentation in the fourth column, where due to the dense ship images, the YOLOv5 algorithm with baseline training did not detect the harbour like RSI-YOLO.

In the third column of Figure 14, it can be observed that the confidence values of the predicted results for baseball field, basketball court, and tennis court targets using RSI-YOLO were all greater than those of the original YOLOv5 algorithm, indicating that the RSI-YOLO model trained on the DOTA dataset has strong feature extraction capabilities. In the fifth column of Figure 14, the comparison of the algorithm’s detection confidence results on the bridge target can also prove that RSI-YOLO has better detection effect than YOLOv5 algorithm with baseline training.

2.DIOR Dataset

The DIOR [33] dataset was also used to validate the effectiveness of the algorithmic improvements. This dataset contains 23,463 images and 192,472 targets with an image size of 800 × 800 pixels, and there are 20 types of targets, namely, airplanes, airports, baseball fields, basketball courts, bridges, chimneys, dams, highway service areas, highway toll booths, golf courses, ground athletic fields, ports, overpasses, ships, stadiums, storage tanks, tennis courts, train stations, vehicles, and wind turbines. The training took a total of 8 h and 27 min. The $AP_{0.95}^{0.5}$ and $AP_{0.5}$ of all categories of the RSI-YOLO were 79.4% and 57.2%, respectively. The detection results for all categories in the dataset are shown in Table 6.

After verification, the detection performance of the trained model using RSI-YOLO was found to be better than that of the original YOLOv5 trained model, which is shown in Figure 15. When comparing the model trained using the baseline YOLOv5s network with that trained using RSI-YOLO, some missed and false detection cases were noted. In the first row of the detection results, for the three aircraft targets on the left, the detection confidence values for the model trained using RSI-YOLO were 0.74, 0.88, and 0.54, whereas for the baseline model the detection confidence values were 0.19, 0.18, and 0.18, which are relatively low. Furthermore, there was one aircraft target on the right that could not be detected by the baseline model, resulting in a missed detection. In the second row of the detection results, the baseline model detected the sidewalk as a bridge, whereas RSI-YOLO did not present a false detection issue. Therefore, RSI-YOLO also displayed good robustness on the DIOR dataset.

3.NWPU VHR-10 Dataset

In addition to the DOTA and DIOR datasets mentioned above, the NWPU VHR-10 [34] dataset was used to validate algorithmic improvements. Compared to the previous two datasets, the NWPU VHR-10 dataset consists of a relatively small set of 650 images containing targets and 150 images without targets. The dataset includes a total of 10 categories of targets, namely, airplanes, ships, oil tanks, baseball fields, tennis courts, basketball courts, athletic fields, ports, bridges, and vehicles. The improved YOLOv5 algorithm was trained on the NWPU VHR-10 dataset. The training took a total of 24 min. Because the dataset is very small compared to DOTA and DIOR, the training time is also faster. The $AP_{0.95}^{0.5}$ and $AP_{0.5}$ for all categories of RSI-YOLO were 95.8% and 57.9%, respectively. The detection results for all categories in the dataset are shown in Table 7.

Some classes such as “airplane” and “tank” achieved nearly 100% $AP_{0.5}$ on baseline training, which is attributed to the small size of the dataset. However, the effectiveness of the algorithm can still be validated by comparing the results in other classes. Several representative target categories were selected, and the results are shown in Table 8. RSI-YOLO improved most of the metrics, with the precision of the “tennis court” category increasing by 6.9%, and $AP_{0.5}$ and $AP_{0.95}^{0.5}$ increasing by 1.7% and 3.9%, respectively. The precision of the “car” category increased by 1.2%, with $AP_{0.5}$ and $AP_{0.95}^{0.5}$ increasing by 2.2% and 1.6%, respectively. $AP_{0.5}$ and $AP_{0.95}^{0.5}$ of the “bridge” category increased by 7.8% and 1.3%, respectively.

4.MAR20 Dataset

Unlike most remote sensing image datasets that contain a variety of different targets such as vehicles, airplanes, and ships, among others, MAR20 [35] is a military aircraft target recognition dataset that provides fine-grained model information. The dataset consists of 3842 high-resolution remote sensing images collected from 60 military airports, including 20 different types of airplanes, denoted by A1–A20. The image size for most of the dataset is 800 × 800 pixels.

Partial detection results and comparisons with original photos are shown in Figure 16. The training took a total of 2 h and 41 min. RSI-YOLO achieved good detection results on different models of this type of aircraft target. 

To compare RSI-YOLO with the original YOLOv5s, several representative target categories were selected, as shown in Figure 17. For this dataset, both initial YOLOv5s and RSI-YOLO achieved good performance in $AP_{0.5}$, with values of approximately 98% and 99%, respectively. The values of this index did not change for “A12” and “A20”, and there is little difference between other categories of $AP_{0.5}$. However, the $AP_{0.95}^{0.5}$ index baseline displayed a better performance effect for RSI-YOLO. Specifically, the $AP_{0.95}^{0.5}$ of “A2” category increased by 4.6%; that of “A9” category increased by 3.6%; and that of “A20” category increased by 3%. These results show that RSI-YOLO provides an improvement over the original YOLOv5s.

### 5.4. Ablation Experiments

To investigate the effectiveness of different improvement methods for algorithm optimisation, ablation experiments were conducted on the DOTA dataset. The YOLOv5 algorithm was separately modified by adding a CBAM attention mechanism, Bi-FPN, small object detection layer, and EIOU loss function. The results of training the YOLOv5s model with each modification are shown in Table 9, where the check mark “√” indicates improvement. It was observed that most of the detection evaluation metrics of the algorithm improved after the addition of each module. Specifically, after the addition of the CBAM module, $AP_{0.5}$ and $AP_{0.95}^{0.5}$ increased by 1.0% and 0.9%, respectively, whereas after the addition of the Bi-FPN module, $AP_{0.5}$ and $AP_{0.95}^{0.5}$ both increased by 1.2%. Although there was no change in the $AP_{0.5}$ value after the addition of the Bi-FPN module, $AP_{0.95}^{0.5}$ increased by 1.5% after the addition of the small object detection layer. Changing the loss function to EIOU increased $AP_{0.5}$ and $AP_{0.95}^{0.5}$ by 0.2% and 1.4%, respectively. After combining the different improvement methods, the combined network model increased $AP_{0.5}$ by 1.3% and $AP_{0.95}^{0.5}$ by 1.6%, achieving the highest performance in the experiment, as expected.

### 5.5. Comparison with Other Object Detection Algorithms

To further verify the effectiveness of the improvement introduced into the YOLOv5s algorithm, RSI-YOLO was compared with other excellent object detection algorithms on the DIOR, NWPU VHR-10 (referred to as VHR-10 in the table below), and MAR20 datasets.

Figure 18 presents partial experimental results on the DIOR dataset, comparing the experimental results of RSI-YOLO, Faster R-CNN, and SSD. Although the precision index of “athletic field” was lower than that of SSD, there was an improvement in $AP_{0.5}$ by 6.4% for RSI-YOLO. For the “basketball” court category, the $AP_{0.5}$ improved by 10.9% and 9.0% for RSI-YOLO and Faster R-CNN, respectively. Moreover, the overall category’s $AP_{0.5}$ increased by 8.5% and 5.8% for RSI-YOLO and Faster R-CNN, respectively.

Part of the experimental results on the MAR20 dataset is presented in Figure 19. Comparing Faster R-CNN and SSD, the $AP_{0.5}$ of category “A13” increased by 11.4% and 3.5%, respectively; the $AP_{0.5}$ of category “A15” increased by 16.4% and 20.4%, respectively; and that of the overall category increased by 2.4% and 3.3%, respectively. For other accuracy and recall rate indexes on this dataset, most values of RSI-YOLO were higher than those of the other two algorithms. Based on the experimental results of RSI-YOLO compared with the other two algorithms, the RSI-YOLO algorithm exhibited superior performance.

Part of the experimental results on the NWPU-VHR 10 dataset is shown in Figure 20. The experimental results of RSI-YOLO, Faster R-CNN, and SSD were compared. The precision of RSI-YOLO was lower than that of the SSD algorithm; however, in terms of $AP_{0.5}$, RSI-YOLO was better than the SSD algorithm. Specifically, $AP_{0.5}$ of the “ship” category increased by 7.2%. The total category of $AP_{0.5}$ increased by 16.6%. Compared with Faster R-CNN, the $AP_{0.5}$ of the “bridge” and total category were also improved by 4.3% and 10%, respectively. The above experimental results show that, compared with other target detection algorithms, the improved algorithm exhibited better detection ability in remote sensing image target detection tasks.

## 6. Conclusions

This study focused on the task of object detection, specifically on the application of remote sensing image object detection, exploring issues related to small target sizes and improving algorithm performance metrics. The feature extraction ability of the original YOLOv5 algorithm was improved by introducing channel attention and spatial attention modules, as well as by modifying the PAN feature extraction structure of the network’s neck section into a weighted Bi-FPN structure, thereby achieving more efficient and rich feature fusion. To address the problem of small target size in remote sensing image object detection, a small target detection layer was added to the network structure to improve the robustness of the network model. Additionally, to further improve the algorithm, the EIOU loss function was incorporated to address the localisation loss function limitations in YOLOv5. This was done to enhance the network’s convergence during training, resulting in an improvement in detection accuracy. The experimental results show that the proposed RSI-YOLO effectively improved detection accuracy and recall compared with YOLOv5 and other typical object detection algorithms.

Although the proposed RSI-YOLO is effective in detecting small targets with improved accuracy, the results suggest that there is still potential for enhancing its average precision on a variety of datasets. Additionally, the proposed RSI-YOLO has achieved remarkable results in improving the accuracy, but the number of parameters in the model increases due to the addition of modules and detection layers, which leads to the increase in FLOPs (floating point operations), so the complexity of the model is higher. When applied in engineering fields, network model parameters and model size are often constrained. Hence, future research can focus on further improving small target detection accuracy. Additional datasets can be used to enrich the network training set, obtaining training samples for data augmentation. For lightweight networks, model pruning and knowledge distillation can be considered to decrease the network model’s size and speed up the detection process while maintaining high detection accuracy.

## Figures and Tables

**Figure 1 sensors-23-06414-f001:**
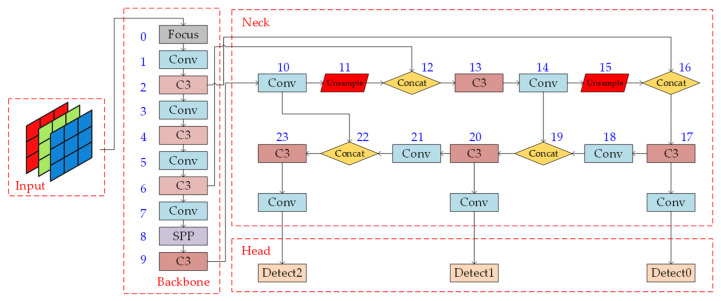
YOLOv5s network architecture.

**Figure 2 sensors-23-06414-f002:**
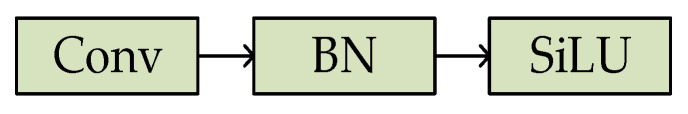
Conv module.

**Figure 3 sensors-23-06414-f003:**
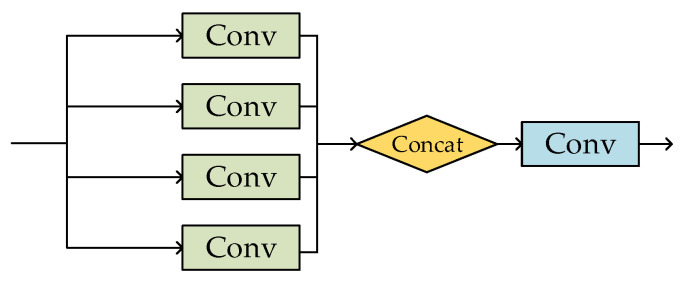
Focus module.

**Figure 4 sensors-23-06414-f004:**
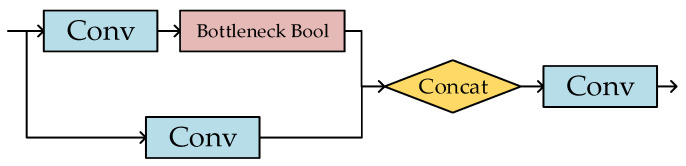
C3 module.

**Figure 5 sensors-23-06414-f005:**
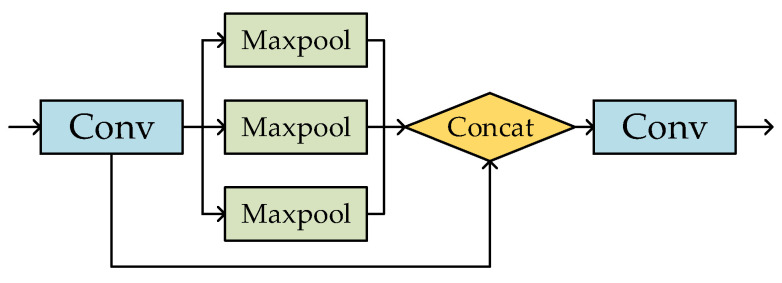
SPP module.

**Figure 6 sensors-23-06414-f006:**
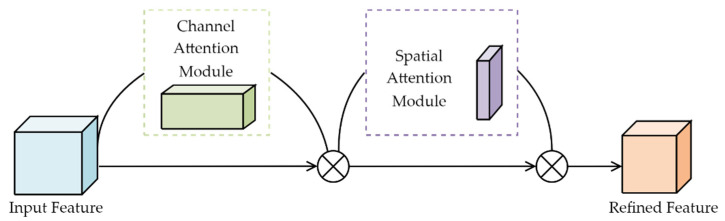
CBAM attention mechanism.

**Figure 7 sensors-23-06414-f007:**
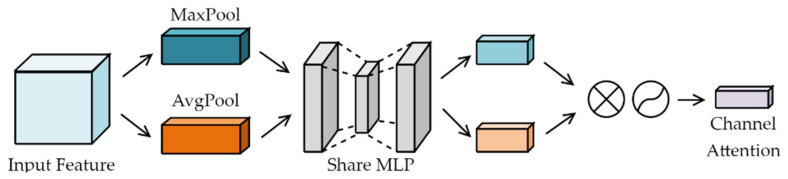
Channel attention module.

**Figure 8 sensors-23-06414-f008:**
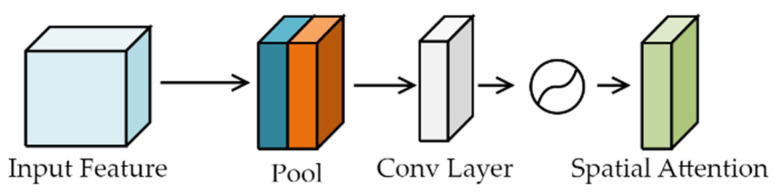
Spatial attention module.

**Figure 9 sensors-23-06414-f009:**
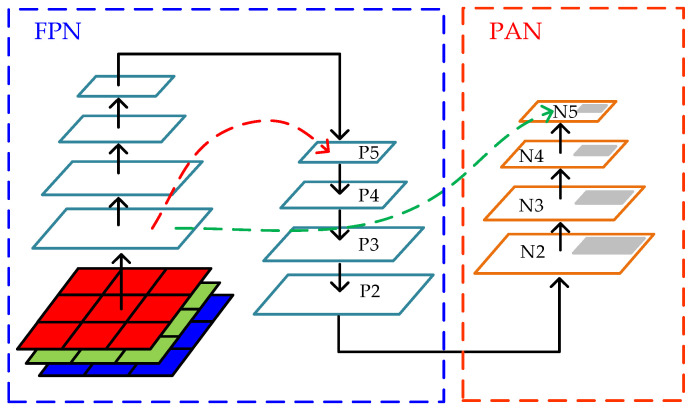
FPN+PAN architecture diagram. The blue box is FPN and the red box is PAN, the red dotted arrow and green dotted arrow are featuring fusion of different scales.

**Figure 10 sensors-23-06414-f010:**
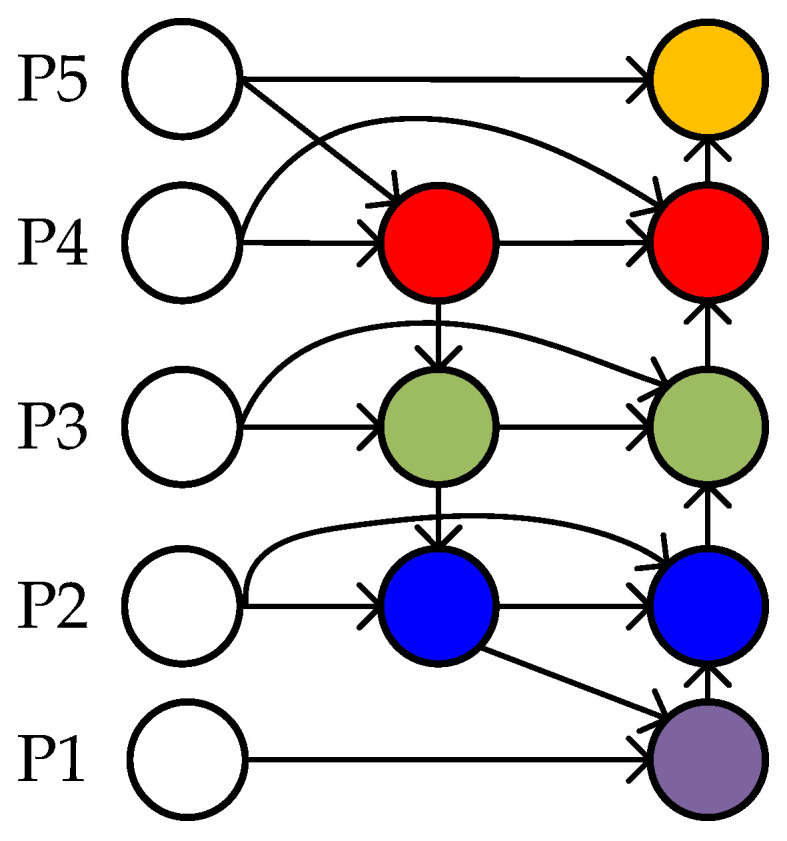
Bi-FPN architecture. Different colours circles represent the features of multi-scale fusion.

**Figure 11 sensors-23-06414-f011:**
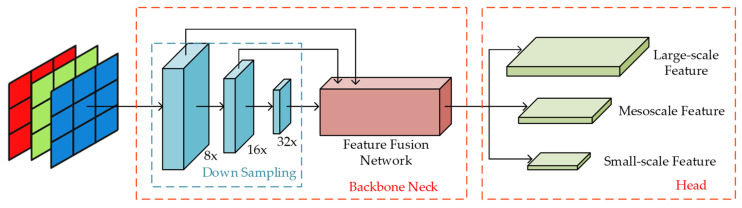
Feature extraction process in YOLOv5. Blue box is downsampling layer, red box on the left is backbone neck and red box on the right is head layer.

**Figure 12 sensors-23-06414-f012:**
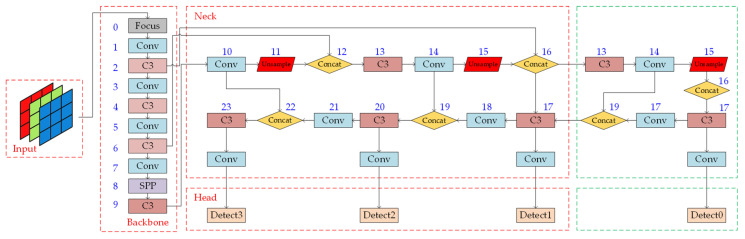
Network architecture with the small object detection layer added.

**Figure 13 sensors-23-06414-f013:**
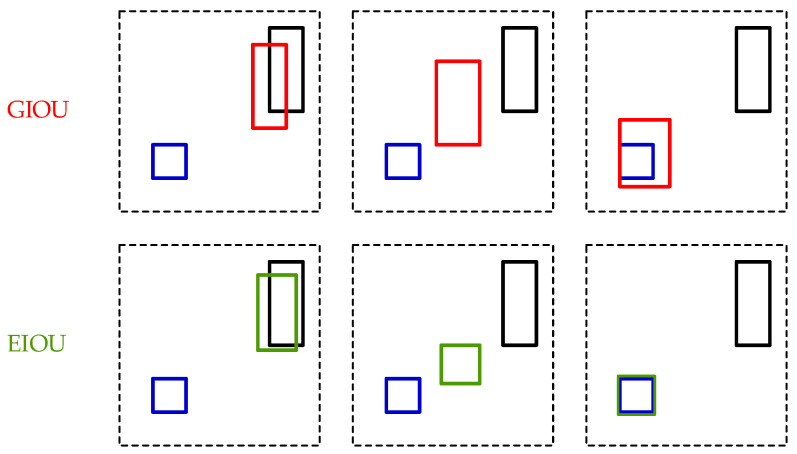
Comparison of iterative process for box prediction using CIOU and EIOU losses. The blue boxes are the true boxes, and the black boxes are the initial predicted boxes. The red and green boxes are the predicted boxes in the convergence process respectively.

**Figure 14 sensors-23-06414-f014:**
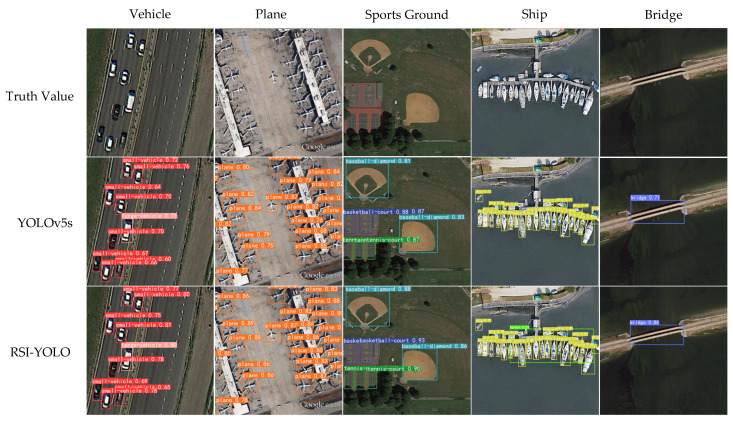
Example comparison of YOLOv5s and RSI-YOLO detection results in the DOTA dataset.

**Figure 15 sensors-23-06414-f015:**
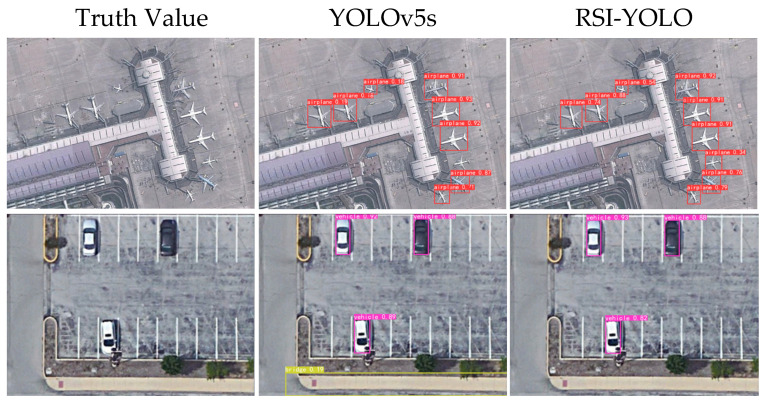
Example comparison of YOLOv5s and RSI-YOLO detection results in the DIOR dataset.

**Figure 16 sensors-23-06414-f016:**
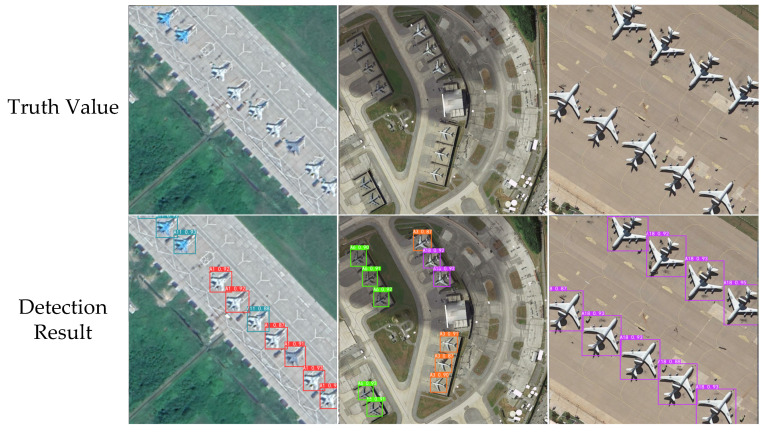
Detection results of YOLOv5s and RSI-YOLO in the MAR20 dataset.

**Figure 17 sensors-23-06414-f017:**
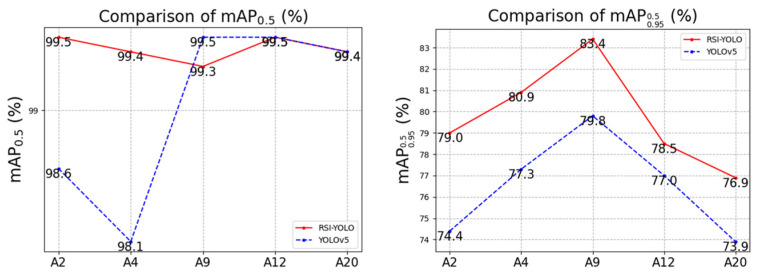
Comparison of RSI-YOLO and YOLOv5s in the MAR20 dataset.

**Figure 18 sensors-23-06414-f018:**
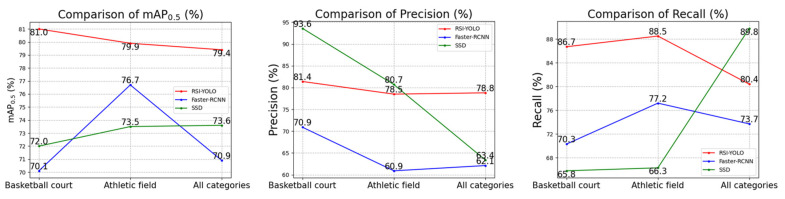
Comparison of RSI-YOLO, Faster R-CNN, and SSD in the DIOR dataset.

**Figure 19 sensors-23-06414-f019:**
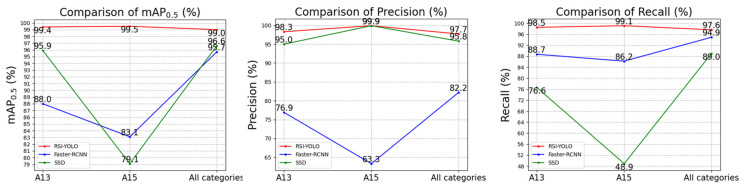
Comparison of RSI-YOLO, Faster R-CNN, and SSD in the MAR20 dataset.

**Figure 20 sensors-23-06414-f020:**
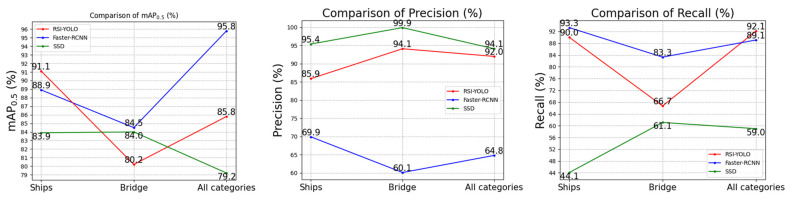
Comparison of RSI-YOLO, Faster R-CNN, and SSD in the NWPU-VHR 10 dataset.

**Table 1 sensors-23-06414-t001:** Configuration of the training environment for the object detection experiment.

Configuration	Version/Model
CPU	Intel(R) Xeon(R) E5-2680 v4@2.40 GHz
Memory	20 GB
GPU	NVIDIA GeForce RTX 3060, 12 GB
Operating system	Ubuntu 18.04
Language	Python 3.8
Acceleration environment	CUDA 11.4
Deep learning framework	Pytorch 1.8

**Table 2 sensors-23-06414-t002:** The configuration of training parameters for the object detection experiment.

Parameter Name	Value
Training epochs	120
Training batch size	32
Number of threads	4
Warmup epochs	3

**Table 3 sensors-23-06414-t003:** The composition of the evaluation metrics for object detection.

Evaluation Metrics		Prediction	
		Positive	Negative
Actual detection	True	*TP*	*FN*
	False	*FP*	*TN*

**Table 4 sensors-23-06414-t004:** Detection results of the proposed improved algorithm.

Categories	P/%	R/%	$AP_{0.5}$/%	$AP_{0.95}^{0.5}$/%
All categories	74.9	69.5	71.2	45.3
Small vehicles	59.8	72.3	67.1	42.0
Baseball diamond	72.7	73.2	75.8	44.2
Large vehicles	83.3	84.4	86.6	63.9
Plane	92.4	88.8	92.2	67.8
Storage tank	89.1	67.4	77.0	45.1
Ship	87.6	86.5	88.2	62.3
Harbour	83.1	82.4	83.9	46.7
Ground track field	74.8	51.3	60.3	33.6
Soccer field	67.2	50.3	54.8	36.6
Tennis court	92.6	89.2	92.5	82.3
Swimming pool	66.9	68.0	62.5	26.1
Roundabout	75.7	56.3	62.0	30.1
Basketball court	65.9	62.0	64.3	46.7
Bridge	64.9	48.8	47.9	20.9
Helicopter	49.7	61.2	53.2	30.9

**Table 5 sensors-23-06414-t005:** Data comparison between RSI-YOLO and YOLOv5s.

Algorithm	Categories	P/%	R/%	$AP_{0.5}$/%	$AP_{0.95}^{0.5}$/%
YOLOv5s	Aircraft	92.2	87.6	91.1	65.5
	Ships	84.7	84.3	86.9	59.5
	Small vehicles	66.6	66.7	68.2	39.5
	All categories	79.0	64.9	69.9	43.7
RSI-YOLO	Aircraft	92.4	88.8	92.2	67.8
	Ships	87.6	86.5	88.2	62.3
	Small vehicles	59.8	72.3	67.1	42.0
	All categories	74.9	69.5	71.2	45.3

**Table 6 sensors-23-06414-t006:** Detection results of RSI-YOLO.

Categories	P/%	R/%	$AP_{0.5}$/%	$AP_{0.95}^{0.5}$/%
All categories	78.8	80.4	79.4	57.2
Aircraft	87.4	99.9	89.9	81.3
Airport	67.9	54.9	67.6	39.1
Baseball field	87.5	98.3	92.4	82.1
Basketball court	81.4	86.7	81.1	68.9
Bridge	70.6	58.2	57.9	35.1
Chimney	92.1	96.2	94.9	84.9
Dam	71.5	66.7	67.0	34.0
Highway rest area	81.6	85.2	83.1	54.3
Highway toll station	93.7	83.1	87.9	70.0
Golf course	65.8	60.6	66.7	38.0
Ground athletics field	78.5	88.5	79.9	64.1
Harbour	75.2	71.1	75.1	44.8
Overpass	80.2	70.3	74.5	51.1
Ship	80.8	96.8	89.3	58.7
Stadium	87.1	82.4	91.5	67.3
Tank	67.5	88.4	81.6	70.1
Tennis court	84.3	93.9	87.7	80.5
Train station	75.5	64.9	66.3	28.9
Vehicle	70.5	74.6	69.3	45.0
Windmill	77.6	86.9	83.8	45.4

**Table 7 sensors-23-06414-t007:** Detection results for RSI-YOLO.

Categories	P/%	R/%	$AP_{0.5}$/%	$AP_{0.95}^{0.5}$/%
All categories	92.0	92.1	95.8	57.9
Aircraft	96.0	99.9	99.5	64.9
Ship	85.9	90.0	88.9	51.6
Tank	94.5	99.9	99.5	51.6
Baseball field	96.8	92.2	96.7	68.0
Tennis court	95.2	97.4	98.9	59.5
Basketball court	84.3	89.4	96.4	52.4
Athletics field	89.9	99.9	99.5	83.0
Harbour	91.3	90.3	97.1	58.5
Bridge	94.1	66.7	84.5	32.1
Car	92.1	95.1	96.7	56.8

**Table 8 sensors-23-06414-t008:** Data comparison between the RSI-YOLO and YOLOv5s.

Algorithm	Categories	P/%	R/%	$AP_{0.5}$/%	$AP_{0.95}^{0.5}$/%
YOLOv5s	Tennis court	88.3	96.9	97.2	55.6
	Car	90.9	85.2	94.5	55.2
	Bridge	88.9	66.6	76.7	29.7
	All categories	93.4	90.2	95.1	56.6
RSI-YOLO	Tennis court	95.2	97.4	98.9	59.5
	Car	92.1	95.1	96.7	56.8
	Bridge	94.1	66.7	84.5	32.1
	All categories	92.0	92.1	95.8	57.9

**Table 9 sensors-23-06414-t009:** Detection metrics of the ablation experiments.

Algorithm	CBAM	Bi-FPN	Small Object Detection Layer	EIOU	$AP_{0.5}$/%	$AP_{0.95}^{0.5}$/%
YOLOv5s					69.9	43.7
	√				70.9	44.6
		√			71.1	44.9
			√		71.1	45.2
				√	70.1	45.1
	√	√	√	√	71.2	45.3

## Data Availability

The data presented in this study are available on request from the corresponding author after obtaining permission of an authorized person.

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
