# Peer review of "RSI-YOLO: Object Detection Method for Remote Sensing Images Based on Improved YOLO"

_sensors, 2023, doi:10.3390/s23146414_

Round 1

Reviewer 1 Report

In general, the structure of the manuscript is good. It is easy to navigate in the manuscript. Unfortunately, the introduction and related work sections are very short. In the introduction, more background information could be provided. The related work section is very short and not written well. The number of cited papers is low. Moreover, coherence is also low. It is rather an unstructured pile of different methods. In Section 3, figure captions should be self-contained and not so short. Loss function is not explained. An explanation, which reveals some information about the motivation of the choices of different terms, would be good and helpful for readers. Same for Section 4. 

Reviewer 2 Report

RSI-YOLO: Object Detection Method for Remote Sensing Images Based on Improved YOLO

Thee present paper focuses on a topic which is not so relevant for the journal. Indeed the work is mainly on the analysis of an existing dataset with some advanced algorithm. Thus this work is apparently better suited for Remote Sensing or for other journals, and only in a limited way for Sensors journal. 

Regarding references I would recommen the authors to add some new recent citations. 

The experimental part is mainly based on DOTA dataset: it would be important to better classify the dataset, including information on different images, amount of distorted images, presence of different distortions (light, dimensions, obstructions, clouds, loweer resolution,...) 

The improvement allowed by the new algorithm is in the point of view of the referee not so determinant if compared to other traditional algorithms.

The work would be more impactnig if additional algorithms are added in the comparison. Also, some other parameters should be considered to better understand the relevance of the new algorithm in terms of lower investigation time, lower costs,... 

It is not clear how ground truth was achieved. If naked eye was used to classify images, authors should consider the possible errors of human eye for objects identification and classification. 

Some Google Earth images are in the crossing position between two different patches, and are characterized by different lighting, resolutions and colors. Authors should consider how the performance of the developed algorithm works (decay?) under this (common) changing conditions. 

English level in general readable

Reviewer 3 Report

The authors have presented an intersting approach called RSI-YOLO for object detection from remote sensing image, based on YOLOv5. The experimental results are satisfactory, but I have several comments that could enhance the quality of the paper:

1- In the introduction, it would be beneficial to include supporting statements explaining why deep learning is advantageous for remote sensing object detection. Please refer to the following recent citations for relevant information:

https://doi.org/10.1007/s12145-022-00885-6

https://doi.org/10.1109/TGRS.2018.2863224

2- Please provide a comprehensive list of the novelties of your proposed method in the final part of the introduction. This will help readers understand the unique contributions of your work.

3- Utilize clear legends for the structures presented in the methodology sections. This will assist readers in comprehending the components of your method.

4- Include the computation time required for training the network. This information is essential for evaluating the efficiency and feasibility of your approach.

5- Since your method focuses on target detection, it would be valuable to showcase more examples of detected objects rather than providing a single comprehensive comparison as shown in Figure 11. You could add additional figures with a multitude of objects that can be detected by your proposed method. Please ensure that each figure is appropriately labeled with the correct names of the objects. Readers expect to see a diverse range of examples in a paper about object detection.

6- To provide a well-rounded analysis, it would be beneficial to include figures illustrating the failures or limitations of the proposed method. Discussing these limitations will open the door for further research and exploration in the field.

Round 2

Reviewer 1 Report

I think the manuscript can be accepted after a minor revision. The figure captions are very short and not informative. More detailed and illustrative figure captions are required. Figure 15 and 16 should be larger.

Reviewer 2 Report

The paper has been improved in agreement with my comments

English in general fine

Reviewer 3 Report

The authors well addressed my comments. 
